# An Edge-Filtered Optical Fiber Interrogator for Thermoplastic Polymer Analysis

**DOI:** 10.3390/s23063300

**Published:** 2023-03-21

**Authors:** Vincent Backmann, Felix Dorner, Klaus Drechsler

**Affiliations:** Chair of Carbon Composites, TUM School of Engineering and Design, Technical University of Munich, 85748 Garching, Germany

**Keywords:** Fresnel reflectometer, refractive index, optical fiber interrogator, coefficient of thermal expansion, thermodynamic polymer analysis, glass transition temperature, crystallization temperature

## Abstract

The present paper deals with the determination of thermodynamic quantities of thermoplastic polymers by using an optical fiber interrogator. Typically, laboratory methods such as differential scanning calorimetry (DSC) or thermomechanical analysis (TMA) are a reliable state-of-the-art option for thermal polymer analysis. The related laboratory commodities for such methods are of high cost and are impractical for field applications. In this work, an edge-filter-based optical fiber interrogator, which was originally developed to detect the reflection spectrum of fiber Bragg grating sensors, is utilized for the detection of the boundary reflection intensities of the cleaved end of a standard telecommunication optical fiber (SMF28e). By means of the Fresnel equations, the temperature-dependent refractive index of thermoplastic polymer materials is measured. Demonstrated with the amorphous thermoplastic polymers polyetherimide (PEI) and polyethersulfone (PES), an alternative to DSC and TMA is presented as the glass transition temperatures and coefficients of thermal expansion are derived. A DSC alternative in the semi-crystalline polymer analysis with the absence of a crystal structure is shown as the melting temperature and cooling-rate-dependent crystallization temperatures of polyether ether ketone (PEEK) are detected. The proposed method shows that thermal thermoplastic analysis can be performed with a flexible, low-cost and multipurpose device.

## 1. Introduction

Thermal analysis of polymer materials can be performed by using well-established analysis techniques such as differential scanning calorimetry (DSC) and thermomechanical analysis (TMA). However, these methods include complex and high-cost devices that are designed only for laboratory use. Therefore, there is a demand for low-cost, multipurpose equipment that can serve the same purposes with acceptable sensitivity and potential field use. For example, strain gauges are a TMA alternative for measuring the coefficient of thermal expansion (CTE) of solid materials [1], and dielectric sensors can be employed for real-time monitoring of the degree of curing in fiber-reinforced thermoset composites [2]. An alternative approach that is insensitive to electromagnetic interference and conductors are optical fiber sensors [3]. They are often applied as intrinsic sensors such that the sensor is the optical fiber itself. In addition, such sensors are relatively small, making it possible to embed them into fiber-reinforced polymer composites without compromising structural properties [4,5]. With this approach, strain and temperature measurements have been performed either in distributed [5,6] or quasi-distributed setups by means of fiber Bragg gratings and their multiplexing capabilities [7].

By utilizing the boundary reflection intensity of an optical fiber tip embedded into a material, the refractive index can be derived via the Fresnel equations (Fresnel reflectometer refractive index sensors, which are referred to herein as FRS). It has been shown that mixing ratios can be detected via the refractive index [8] and the degree of cure in thermoset-based polymers [9,10,11,12,13,14,15]. A direct comparison of refractive indices and dielectric cure sensors revealed good agreement for cure monitoring of a thermoset-based resin [16]. Early detection setups to measure the boundary reflection intensity were equipped with one light source and a single photodiode [12]. To expand the functionality of the setups, a device with two center wavelengths at 1310 nm and 1550 nm to simultaneously monitor fiber Bragg gratings and Fresnel reflectometer sensors has been developed [17]. A further enhancement for glass transition and gel-point monitoring of thermoset resin was performed in [18]. This principle was adapted and employed to monitor glass-fiber-reinforced thermosets [19].

With multiple publications focusing on the application of FRS to monitor-curing reactions of thermoset-based resins in which the molecular chains crosslink, thermoplastic polymers are bonded by intermolecular forces. Below the glass transition temperature, amorphous polymers are considered to be solid/glassy; above this temperature, the polymers are in a liquid/rubbery state due to movements of the molecular chains [20]. The temperature-dependent refractive indices of the amorphous polymers polycarbonate (PC), poly(methyl methacrylate) (PMMA) and polystyrene (PS) have been found to be linear within a polymer material phase. Each of these phases is characterized by a different slope [21,22]. This behavior was reproduced in an FRS setup. It was demonstrated that the determination of the glass transition temperature in such an amorphous thermoplastic polymer is obtained by calculating the intersection point of the fitted lines of the glassy and rubbery phases. Good agreement with DSC measurements has been found [12].

Despite the ability to detect the glass transition temperature of amorphous thermoplastic polymers by using the FRS, the close relation between refractive index and density as expressed by the Lorentz–Lorenz model is a potential enhancement of this type of refractive index sensors toward density-related thermodynamic quantities. In addition, the other thermoplastic material group, namely, the semi-crystalline polymers, have not been investigated by using temperature-dependent FRS measurements. This group of polymers consists of a glass transition, melting and crystallization temperature [23]. The latter defines the growth of a district crystal structure subjected to the thermal processing conditions. These crystals consist of a different density to the non-crystalline regions, which results in a change in bulk density [24] and, thus, the refractive index. From that, other density-related properties and thermodynamic quantities may potentially be detected. However, with the presence of crystals, a distortion of the refractive index measurements is a potential source of error when effects of birefringence are neglected.

Most commonly, laboratory devices such as DSC are the industrial standard in the detection of transition processes and the degree of crystallinity of thermoplastic of polymers. TMA is a widely accepted CTE measurement device. The potential low-cost alternative for the detection of the transition temperatures and CTE by means of the refractive index are edge-filtered optical fiber interrogators, which were originally developed as detection units for fiber Bragg grating sensors. These devices split the reflection signal into two optical paths, with one of these equipped with an intensity detector and the other consisting of the edge filter and a photodiode [25]. Oelhafen et al. employed such a device for refractive index measurements by using a cleaved end of a pigtail [13]. A calibration procedure using fluids of known refractive indices, similar to [12], revealed an offset in the range of $10^{-3}$ and an uncertainty of $10^{-4}$ in refractive index. The capabilities of this method were demonstrated by means of temperature-modulated optical refractometry [13], cure measurements and an empirical relation of the viscosity to the refractive index of a thermoset-based resin [26].

The present work aims to qualify the edge-filter-based optical fiber interrogator utilized in [13] for the detection of thermodynamic quantities of thermoplastic polymers via the refractive index. For that, the temperature-dependent refractive indices of two amorphous thermoplastics, namely, polyetherimide (PEI) and polyethersulfone (PES), are investigated. Neither has been previously analyzed via this method, and, therefore, they broaden the spectrum of polymer analysis. The relation from [27] is applied to obtain the CTE from the temperature-dependent refractive index via the Lorentz–Lorenz model. The glass transition temperatures and the CTE directly detected from the refractive index of the amorphous thermoplastic polymers are successfully verified. To address the group of semi-crystalline thermoplastic polymers, polyether ether ketone (PEEK) is further analyzed. With the absence of crystals, the melting temperature and the cooling-rate-dependent crystallization temperatures derived via FRS are both well replicable via DSC. With the presence of crystals, no acceptable agreement of PEEK is found in the correlation of degree of crystallinity and the CTE via the Lorentz–Lorenz equation. Although crystals are present in the range of the glass transition temperature, it is detected with a weak sensor response signal. The method proposed in this work presents a low-cost alternative to DSC and TMA in the analysis of amorphous thermoplastic polymers. The temperature-dependent refractive index analysis of semi-crystalline polymers is limited to the detection of transition temperatures with the absence of crystals indicating a DSC alternative.

## 2. Methods and Experimental Setup

### 2.1. Optical Fiber Fresnel Reflectometer—Refractive Index Measurement

The edge-filter-based optical fiber interrogator developed for FBG sensor detection by fos4X GmbH, Munich, Germany (now Polytech A/S, Bramming, Denmark) was utilized for refractive index measurements. It consists of an amplified spontaneous emission light source emitting a center wavelength of circa 1550 nm. From that, the light is split by a wavelength-division multiplexer and guided into each of the four channels to the connectors of the external sensor attachments. In every optical channel, the reflected sensor signal is split into two paths. In one path, a photodiode directly measures the reflected signal, whereas in the other path, another photodiode detects the signal after passing through an edge filter. For further information and the schematic device setup, the reader can refer to [26]. In this work, only the reflected intensity signal from the unfiltered path was used for refractive index measurements.

The relation of the reflection intensity detected via the optical fiber interrogator to measure the refractive index is described by the Fresnel equations, which correlate the boundary reflection of the incident light of two optical transparent materials under contact and the refractive indices of these materials. Assuming normal light incidence and flat boundary surfaces of a cleaved straight optical fiber tip embedded into a media (Figure 1), the Fresnel equations are simplified to
(1)$$\frac{I_{r,media}}{I_{i}}=\frac{{\left(n_{eff}-n_{media}\right)}^{2}}{{\left(n_{eff}+n_{media}\right)}^{2}}$$
where $I_{r,media}$, $I_{i}$ are the reflected and incident light intensities, and $n_{eff}$, $n_{media}$ are the refractive indices of the optical fiber and its surrounding media, respectively.

In order to create the flat fiber tip, the jackets and acrylic coatings of SMF28e single-mode optical fiber pigtails from Corning Inc. (Corning, New York, NY, USA), with 125 µm claddings and 8.2 µm core diameters, were stripped and cleaved. These pigtails were connected to the optical fiber interrogator via LC/APC connectors. The incident light intensity emitted by the laser light source of the measurement device was unknown. In order to find this parameter, the procedure from [12] was applied. According to this, a reference measurement was carried out in known media while neglecting losses in the optical path. For this purpose, the incident light intensity $I_{i}$ was experimentally determined in ambient air with $n_{air}$ of 1.00027 at 23 °C [28], yielding the corresponding $I_{r,air}$. Then, $I_{i,air}$ was derived by using Equation (1) and substituted into Equation (1). Thus, Equation (1) with $I_{i}$ equal to
$I_{i,air}$ takes the form of
(2)$$\frac{I_{r,media}}{I_{r,air}}=\frac{{\left(n_{eff}-n_{media}\right)}^{2}}{{\left(n_{eff}+n_{media}\right)}^{2}}\;\frac{{\left(n_{eff}+n_{air}\right)}^{2}}{{\left(n_{eff}-n_{air}\right)}^{2}}$$

Solving the quadratic equation (Equation (2)) for the refractive index of the media under investigation results in
(3)$$n_{media}=n_{f}\left(T\right)\cdot \frac{1+a}{1-a}\;\forall \;n_{media}\geq n_{eff}\;with\;a=\frac{n_{eff}-n_{air}}{n_{eff}+n_{air}}\;\sqrt{\frac{I_{r,media}}{I_{r,\;air}}}$$

The influence of the refractive index of the optical fiber to the temperature T is
(4)$$n_{f}\left(T\right)=n_{eff}+TOC\;\left(T-T_{0}\right)$$
where $T_{0}$ is 23 °C, and the effective refractive index $n_{eff}$ is 1.4473 of the SMF28e optical fiber used in [29] with the thermo-optic coefficient TOC of fused silica of $8.7\times 10^{-6}$ [30]. The optical fiber was doped with other elements rather than composed solely of fused silica. The fact that the degree of doping of the optical fiber changes this parameter and its dependency on temperature, as shown in [31,32,33], was neglected in this analysis.

### 2.2. Lorentz–Lorenz Model—Correlation of Refractive Index and Thermal Expansion

The Lorentz–Lorenz equation relates the mass density ρ and refractive index n for optically transparent or translucent media up to at least a certain degree, as shown in the following equation [34]:(5)$$\rho =\frac{1}{r}\;\frac{n^{2}-1}{n^{2}+2}$$
with the specific refraction r remaining unknown. An analysis of this parameter requires additional time-consuming experiments and infrastructure to perform those experiments and diminishes the advantage of a direct detection of the CTE from the refractive index. Müller et al. [27] proposed that temperature-related changes in the refractive index are predominantly related to changes in mass density. Therefore, r is considered constant and independent of temperature influences. When converting density into the thermal expansion of an isotropic material, the CTE is directly related to the refractive index as follows:(6)$$\alpha =\frac{-2n_{0}}{\left(n_{0}{}^{2}+2\right)\left(n_{0}{}^{2}-1\right)}\cdot \Psi$$

Here, $\Psi =\frac{dn}{dT}$ is the derivative to determine the temperature-dependent CTE. A linear change in refractive index over a large range of temperature is defined as $\Psi =\frac{\Delta n}{\Delta T}$.

### 2.3. Experimental Setup

Thermoplastic polymers are of a relatively high viscosity in the liquid state in comparison to thermoset resins, e.g., epoxy resins. To reliably cover the tip of the optical fiber, a vacuum bagging setup was utilized. For that step, the cleaved end of the optical fiber was placed between two thermoplastic films of each polymer under investigation. The surrounding vacuum bagging stacking sequence is illustrated in Figure 2. First, the polymer films with the optical fibers were placed on a polyimide (PI) film laid on the steel plate (Figure 3a). A perforated PI film and glass fabric were then placed on top of this, acting as a breather to ensure airflow within the sample (Figure 3b). A standard hypodermic needle for syringes of medical applications, which was protected with glass fabric at the tip to prevent cuts to other surrounding materials, was placed on top of the glass fabric breather (Figure 3b). Finally, a PI film covered this setup and was secured with the vacuum sealant tape (GS A-800-3G Sealant Tape, Generals Sealants Inc., City of Industry, CA, USA). In order to measure the temperature near the pigtail tips, two thermocouples (TC) of Class 1 type K (GG-KI-30-SLE, OMEGA Engineering, Norwalk, CT, USA) were fed through the sealant tape. The thermocouples have an accuracy range of 1.5 °C and were not subjected to additional calibration procedures. The thermocouple data logger PicoLog tc-08 (Pico Technology Ltd., Cambridgeshire, UK) was used for recording the temperature. Cooling rates exceeding 60 °C/min were measured by using an HBM QuantumX MX840B (Hottinger Brüel & Kjaer GmbH, Darmstadt, Germany) device. A computer was connected to the interrogator and thermocouple data logger to record the measurement data (Figure 4). The entire vacuum bagging setup was the size of a microscopy slide and was placed into the Mettler–Toledo hot stage furnace HS82 (Columbus, OH, USA).

## 3. Amorphous Thermoplastic Polymers: Polyetherimide (PEI), Polyethersulfone (PES)

### 3.1. Experimental Details and Materials

Two amorphous thermoplastic films PEI and PES supplied by LITE GmbH, Gaflenz, Austria, were analyzed. The embedding of the FRS in accordance with the vacuum bagging setup under atmospheric pressure was performed at 290 °C for both polymers. The coverages of the optical fiber tips in the ranges of each of the glass transition temperatures were characterized by a clear change in reflected intensity signal. The experiments were conducted with three heating and cooling cycles of 20 °C/min on each polymer film. Three optical fiber pigtails were embedded into each experimental setup, yielding three corresponding data channels.

### 3.2. Validation: TMA and DSC

The thermal expansion of the polymer films under investigation and the glass transition temperatures were examined with a thermomechanical analyzer TMA 402F1 (Netzsch GmbH & Co. KG, Selb, Germany). In order to investigate transverse isotropic effects of the films, three specimens were cut from each in two perpendicular directions, yielding six specimens for each material. The specimens have rectangular forms with lengths of about 15 mm and widths of about 5 mm. A constant tension of 0.01 N and a heating rate of 1 °C/min were used in the TMA tensile clamp setup.

DSC measurements taken by using a DSC Q200 (TA Instruments, New Castle, DE, USA) with heating rates of 20 °C/min were conducted as other referencing methods to determine the glass transition temperatures. Three specimens of each polymer with masses of circa 7 mg were tested. The accurate mass was estimated for each specimen separately.

### 3.3. Results and Discusion

#### 3.3.1. Refractive Indices

The refractive indices of PEI and PES were calculated from Equation (3) by using the reflected intensity signals measured with the optical fiber interrogator. The dependencies of the resulting refractive indices on the temperatures are shown in Figure 5. With an increase in temperature, the refractive indices of PEI and PES decrease. Two different regions of linear behavior, which are characterized by different slopes separated somewhere around the glass transition temperatures of the material datasheets of 215 °C and 230 °C of PEI and PES, respectively, [35,36] are observed. For PEI, the mean refractive index and standard deviation at 25 °C for all three channels and all cycles used in the experiment is 1.6119 ± 0.0072 (Figure 5a). This is in close agreement with [34], where a refractive index of 1.6165 was reported for a wavelength of 1550 nm at room temperature. The mean standard deviation of the refractive index subjected to all temperatures is 0.0082. For comparison, the mean standard deviation of all single channels that were used in the measurements is lower at 5.5 × 10^−4^ at 25 °C, and throughout all temperatures, it is 7.0 × 10^−4^. In contrast to that, the standard deviation of all three channels of PES is smaller at 0.0021 at 25 °C and 9.9 × 10^−4^ over the entire temperature range. Therefore, the refractive index results are summarized in only one curve (Figure 5b). It is notable that the accuracy of the absolute refractive index estimation is lower compared to the relative change in this parameter from temperature influences. A possible explanation of this behavior is the procedure to determine the incident light intensity of the measurement device by the use of air. This is dependent on the creation of a straight fiber tip with an optical fiber cleaver. Further quality analysis with the cleaved fiber tip, e.g., optical analysis or further calibration procedures as proposed by [12,13], are not performed for reasons of simplicity. This may result in a systematic error of absolute refractive index determination, whereas the relative temperature-dependent change is of higher accuracy.

#### 3.3.2. Coefficients of Thermal Expansion

The CTEs are obtained based on the temperature-dependent refractive indices of both amorphous polymers under investigation. When linear behavior of the temperature-dependent refractive index is considered, the CTE is constant within the selected range of temperatures. For this purpose, straight lines are fitted into each section between 25 °C and 180 °C and between 240 °C and 280 °C, which are below and above the glass transition temperature ranges of both polymers, respectively (Figure 5). Inserting these values into Equation (6) reveals the linear CTEs that are summarized in Table 1.

In order to validate the results obtained from the FRS, the TMA is applied. In contrast to the decreases of refractive indices with rising temperatures, the mean thermal strains derived from the TMA exhibit a linear increase with rising temperatures of PEI and PES (Figure 5). The slopes of the temperature-dependent strain responses are the linear CTE, which is obtained by fitting a straight line between 25 °C and 180 °C for both polymers under investigation. The results of PEI and PES are shown in Table 1. After reaching the glass transition temperature ranges, sharp changes in the strain responses are observed. These contractions in the tensile clamp setup of the TMA may be related to stress relaxation effects from the film production process, as the polymers are transferring from the glassy to rubbery phase, allowing movement of the molecular chains. An estimation of the CTE at these temperatures does not lead to valid results in this tensile setup in the TMA. The requirements of this TMA configuration of the absence of stress relaxation effects in phase transitions and a minimum mechanical material cohesion being in contrast to an optical method using FRS is not essential. A dependency on the cut-out direction from the main film on the results is not observed. The CTEs determined with the TMA below the glass transitions are presented in Table 1. These are in good agreement with those obtained by using the optical fiber method. In addition, the linear CTEs from this work are in close correlation with the material datasheets, which state a CTE of 56 × 10^−6^ °C^−1^ for PEI [35] and 55 × 10^−6^ °C^−1^ for PES [36].

It is generally desirable that only a few measurements of high accuracy are sufficient for detecting the physical parameter of interest. Therefore, the absolute difference between the largest and smallest value of the linear CTE is estimated. For PEI, the difference between these values is of 16.6 × 10^−6^ °C^−1^ for the glassy phase and 13.4 × 10^−6^ °C^−1^ for the rubbery phase. For PES, it is of the same magnitude, namely, of 11.9 × 10^−6^ °C^−1^ and 16.7 × 10^−6^ °C^−1^ for the solid and rubbery states, respectively. Meanwhile, these differences derived from TMA are 9.2 × 10^−6^ °C^−1^ for PEI and 11.1 × 10^−6^ °C^−1^ for PES in the solid phases, which is not significantly smaller in comparison to those obtained from the FRS measurements.

An alternative to the constant representation of the CTE within a temperature range is the consideration of the natural non-linear behavior of this parameter. This is derived by using the local temperature derivatives of the refractive index with $\Psi =\frac{dn}{dT}$ from Equation (6) and a temperature-dependent strain $\frac{d\varepsilon }{dT}$ of the TMA. For each specimen, the derived CTE value consists of a high level of noise due to the derivatives. When a median filter is applied, the noise is reduced. Finally, the filtered data of all CTE values are summarized in a mean value and the corresponding standard deviations. It is shown that the CTEs of both polymers and methods slightly increase with rising temperatures (Figure 6). The TMA and FRS with the applied Lorentz–Lorenz model reveal an acceptable agreement. In the following transition phase, the FRS data reveal large changes of CTEs over relatively small temperature ranges. Finally, the CTEs of the rubbery phases continue without pronounced temperature-related changes. In this state of polymer, the molecular chains of the polymers are considered to be movable. As a result, no stress–strain relation persists in the interface of the optical fiber and polymer. However, during the cooling down from this transition phase into the glassy state, the differences of the mechanical properties of the optical fiber and polymer are potentially a cause of a significant mechanical perturbation as a result of the mechanical interaction of two bonded dissimilar materials, thus influencing the measured refractive index. The extent of this is yet to be determined. No such pronounced deviation is found within the tested temperatures, but it may potentially become detectable at temperatures below those applied in this study.

#### 3.3.3. Glass Transition Temperatures

The polymers under investigation undergo a transition process from a glassy to a rubbery state in the tested temperature ranges. The refractive indices and thermal strains from Figure 5 and the CTEs derived from FRSs of Figure 6 reveal the clear change in the magnitude of the properties. To describe these transition phases, it is practical to further analyze the temperature-dependent CTEs derived from the FRSs in Figure 6. The beginnings and endings are described by the extrapolated onset and offset temperatures, respectively. These are determined by calculating the intersections of the extrapolated linear slopes of the glassy phase with the transition phase (onset) and the transition phase with the rubbery phase (offset). The extrapolated onset temperatures for PEI and PES are 202.6 °C and 211.5 °C, respectively. The extrapolated offset temperatures are found to be 225.6 °C for PEI and 243.5 °C for PES. In comparison to that, the DSC reveals a smaller temperature range of the transition phase. The extrapolated onset temperatures are 213.8 °C for PEI and 223.8 °C for PES. The extrapolated offset temperatures of PEI and PES derived from the DSC are 220.9 °C and 230.5 °C, respectively.

A convenient parameter to describe the occurrence of the phase transition is the glass transition temperature. This temperature is somewhere between the onset and offset temperatures depending on the calculation scheme. To estimate the glass transition temperature, it is straightforward to calculate the intersection points of the two linear curves of the glassy and rubbery phases from Figure 5a,b of the temperature-dependent refractive indices and thermal strain responses from the TMA. The results are shown in Table 2. One drawback of this procedure is the low estimations of the transition temperatures from the refractive indices in comparison to the inflection points of the heat flow response signal derived from the DSC. The intersection points derived from the thermal strain responses of the TMA from Figure 5 reveal another disadvantage of this procedure, which is that of their high dependency on the slopes of both curves. Due to the stress relaxation contractions of the polymer films, the resulting steep slopes lead to higher estimates of the glass transition temperature. These relaxation effects occur randomly distributed throughout the specimens, which can result in a higher standard deviation, as the PES transition temperature shows.

In close accordance to the DSC transition temperatures are the inflection points derived from the CTEs of the FRSs (Figure 6). The offsets between these methods are 3.5 °C and 0.9 °C for PEI and PES, respectively (Table 2). It becomes clear that the results are dependent on the measurement method and data analysis procedure. The glass transition inflection points derived via the CTE from the refractive indices are in good agreement with the DSC results and the material datasheets. These datasheets state 215 °C for PEI and 230 °C for PES derived from the ISO 11357 (DSC) method. 

## 4. Semi-Crystalline Polymer: Polyether Ether Ketone (PEEK)

### 4.1. Experimental Details and Materials

The semi-crystalline polymer film material PEEK (Aptiv 1000, Victrex, Thornton Cleveleys, Lancashire, UK) is equipped with the FRS for refractive index measurement. The integration of the prepared optical fiber pigtails is performed by using the vacuum setup described above. A heating rate of 20 °C/min at atmospheric pressure up to 375 °C is conducted for the embedding process. A clear change in reflected intensity signal is detected around the expected melting temperature range. The experiments are conducted by using three channels of the optical fiber interrogator. Controlled cooling rates of 1 °C/min, 2 °C/min, 5 °C/min, 10 °C/min, and 20 °C/min are performed by using the hot stage. Cooling rates of circa 60 °C/min, 500 °C/min, and 4000 °C/min are reached by opening the furnace, placing the specimen onto an aluminum plate at room temperature and with a bath of isopropyl alcohol with dissolved dry ice reaching a temperature of circa −74 °C. Note that cooling rates above 20 °C/min are uncontrolled.

### 4.2. Validation: DSC and DMA

The validation of the results obtained via the optical fiber experiment and analysis was conducted by corresponding DSC measurements using a DSC Q200 (TA Instruments, New Castle, DE, USA). With a heating rate of 20 °C/min, the degree of crystallinity, glass transition, and melting temperatures were determined in the first heating cycle from each specimen of one corresponding cooling rate. In order to characterize the onset crystallization temperatures, multiple cooling rates were performed up to 20 °C/min, whereas rates above this were conducted by using a DSC 8500 from PerkinElmer, Inc., Waltham, MA, USA. The lids and covers were filled with a polymer film of circa 7 mg each. The mass of each specimen was measured separately to calculate the degree of crystallinity.

In order to determine the CTE of the PEEK polymer film, a DMA Q800 (TA Instruments, New Castle, DE, USA) was utilized for CTE measurements. With an applied constant tensile force of 0.01 N and a heating and cooling rate of 1 °C/min, the device was calibrated by using aluminum for a machine calibration procedure. With additional strain gauges attached to the aluminum specimen, the temperature-dependent CTE was determined according to the proposed method of [1]. A deduction of the temperature-dependent machine offset was determined and compensated for this experimental work. The CTE of the PEEK films was measured in the tensile clamps with a force of 0.01 N by using heating and cooling rates of 1 °C/min for a total of three specimens with lengths of circa 20 mm and widths of 5 mm.

### 4.3. Results and Discussion

#### 4.3.1. Refractive Index

By subjecting the polymer film with the embedded optical fiber pigtail to the heating and cooling cycles, the reflected intensity signals were measured via the edge-filtered optical fiber interrogator. The temperature-dependent refractive index of the polymer including the compensation of the temperature was obtained by using Equation (3). Figure 7a shows that the heating and cooling curves superimposed while decreasing with increasing temperatures. The refractive index response signals separated at a few tens of degrees Celsius below the melting and crystallization temperatures, Tm and Tc, respectively. These temperatures were defined either as intersection points of two straight lines fitted into the refractive index signals, or as the maximum of the temperature derivative, yielding the inflection point (Figure 7b). Below 300 °C, the standard deviation of the refractive index was lower at 0.0062 ± 6.0 × 10^−4^ when compared to those above the transition temperatures at 0.0015 ± 1.0 × 10^−4^. The larger standard deviation in the rubbery phase is a potential result of the presence of stress between the optical fiber and the polymer material due to a mismatch of elasticity and thermal expansion between these two materials under contact. The persistence of bonding between these two different materials is shown by a microsection in Figure 8a. It reveals the presence of micro-cracks surrounding the optical fiber tip. The deviation below both of the solidification temperatures, melting and crystallization temperatures, remains constant, although the residual stresses between these two dissimilar materials (optical fiber and polymer) is expected to increase with a decrease in the temperature. In addition, this effect was not observed in the investigation of the amorphous thermoplastic materials. Therefore, the larger deviation is attributed to the presence of crystals below the crystallization and melting temperatures, which causes a birefringence, as Figure 8b shows in a petrographic microscope examination of a thin section at visible light. The FRS, including the measurement system and design, excludes methods of polarization control and management, yielding larger deviations of the obtained refractive indices. Depending on the thermal history, with the size of a crystal typically ranging from 1 µm to 10 µm, the shape and crystallized content up to circa 40% varies [37]. In addition, the optical properties of a unit cell crystal differ from the orientations and principal polarizations [38]. The refractive index at room temperature with polymer cooling rates below 100 °C/min is 1.664, which is within the values reported by [38], depending on the principal polarization. Applying the setup from this work by using a pigtail of 8.2 µm for a precise refractive index determination is yet to be solved.

#### 4.3.2. Thermal Expansion

The characteristic behavior of the temperature-dependent refractive index derived from the PEEK polymer (Figure 7a) relates somewhat inversely proportionally to the density, as reported by [23]. The close relation of density and CTE can be expressed via the Lorentz–Lorenz equation, yielding the CTE directly from the temperature-dependent refractive index (Equation (6)). The mean value derived from all refractive index measurements is shown in Figure 9. For comparison, the CTEs obtained via the calibrated DMA are in good agreement with [39], but no superposition is found with the FRS. Both results (DMA and FRS) are median-filtered to reduce the noise created from the temperature derivatives. Despite following a comparable pattern, separation increases with temperature. This can be ascribed to the measurement setup of the refractive index by using the simplified Fresnel equations neglecting birefringence and the estimation of the CTE assuming constant specific refraction from the Lorentz–Lorenz relation.

#### 4.3.3. Transition Temperatures

Directly estimating the glass transition temperature of PEEK from the temperature-dependent refractive index (Figure 7a) is unfeasible due to the low change in measurement signal in the expected temperature range. However, a small change in slope is found by using the CTE derived from the refractive index (Figure 9). In this case, it becomes detectable only because the mean value of all measurements are median-filtered. Here, the inflection point is calculated with the result presented in Table 3. Despite this low quality of the refractive index signal, it is in good agreement with the inflection point calculated from the CTE response of DMA. In addition, the inflection point from the DSC heat flow response signal shows good agreement with both of the other methods.

In contrast to this, the melting temperature is characterized by a clear change in the signal of refractive indices (Figure 7a). The estimation of the inflection and intersection temperatures from the Fresnel reflectometer sensor is within a 2 °C range, as shown in Table 3. In comparison to the DSC peak heat flow signal, both of the temperatures are higher estimates with acceptable agreements to the melting temperature derived from DSC.

The beginning of the crystallization process during the cooling of the polymer material can be described by using the onset crystallization temperature. It can be detected by DSC and has been shown to be dependent on the cooling rate (Figure 10) [40]. Figure 11 shows that with an increase in the cooling rate, a retardation of the characteristic change in slope of the refractive index towards lower temperatures is observed. To express this behavior by means of the crystallization temperature, the intersection of two straight lines according to Figure 7b, is determined. The influence of cooling rates ranging from 1 °C/min to 4000 °C/min on the crystallization temperature is shown in Figure 10. By comparing the results to the onset crystallization temperatures in accordance to ISO 11357 derived via DSC, the same trend with a decrease in crystallization temperature as a result of an increase in the cooling rate is observed. In general, the results obtained from the refractive index are of higher estimates than those obtained via DSC at cooling rates of up to 20 °C/min. The uncontrolled rates above are of a higher standard deviation of refractive index. The overestimation is, therefore, not pronounced. Considering the nature of the detection of the crystallization temperature from a different physical parameter (thermal energy (DSC) and refractive index), an acceptable agreement can be concluded.

#### 4.3.4. Degree of Crystallinity

The density is related to the refractive index by means of the Lorentz–Lorenz equation and also to the degree of crystallinity [24,37]. Therefore, a direct detection of the degree of crystallinity via the refractive index by using the FRS is a potentially straightforward solution. For that purpose, the degree of crystallinity of PEEK referring to 130 kJ/kg [40] was performed by using the specimens directly conditioned from each cooling rate of the optical fiber measurements (Figure 12). The degree of crystallinity follows a declining trend with increasing temperatures in accordance to [24]. The results from the FRS reveal no such trend. The refractive index measurements are constant throughout the cooling rate and exhibit a significant change only at cooling rates of 500 °C/min and above. A clear distinction of the refractive index between cooling rates of up to 60 °C/min and above 500 °C/min is not possible. The FRS in contrast to DSC is not suited for the direct detection of the degree of crystallinity of this semi-crystalline material.

## 5. Conclusions

A method of thermal thermoplastic polymer analysis has been proposed and demonstrated. By utilizing an edge-filer-based optical fiber interrogator that was equipped with a cleaved pigtail of a standard telecommunication optical fiber, the temperature-dependent refractive index was derived by means of the Fresnel equations. It is exemplary revealed via the analysis of the amorphous thermoplastic polymer materials PEI and PES, that the method is an alternative to DSC and TMA in glass transition and coefficient of thermal expansion detection. The analysis of the semi-crystalline polymer reveals the limitation, that only the transition temperatures in the absence of crystals can be reliably detected. The proposed method is a low-cost and flexible solution, that is, in their valid filed of application, competitive with standard industrial laboratory devices. 

## Figures and Tables

**Figure 1 sensors-23-03300-f001:**
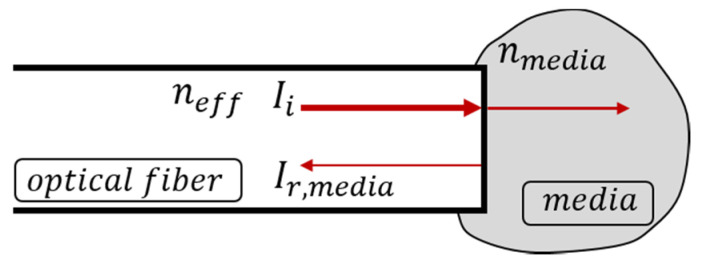
Simplified schematic representation of the Fresnel reflection of the optical fiber and media.

**Figure 2 sensors-23-03300-f002:**
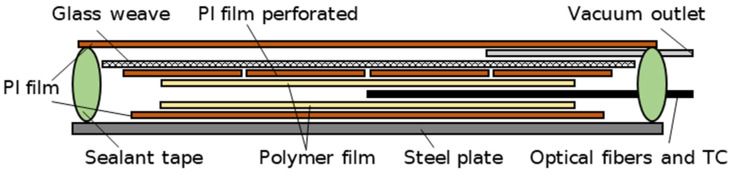
Schematic representation of the experimental vacuum bagging setup. TC refers to the thermocouple.

**Figure 3 sensors-23-03300-f003:**
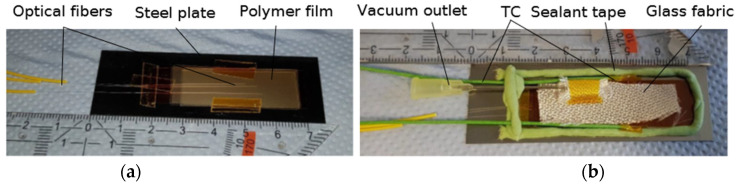
Stages of vacuum bagging setup with the films and the prepared optical fibers (**a**) and before sealing the setup with a PI film (**b**).

**Figure 4 sensors-23-03300-f004:**
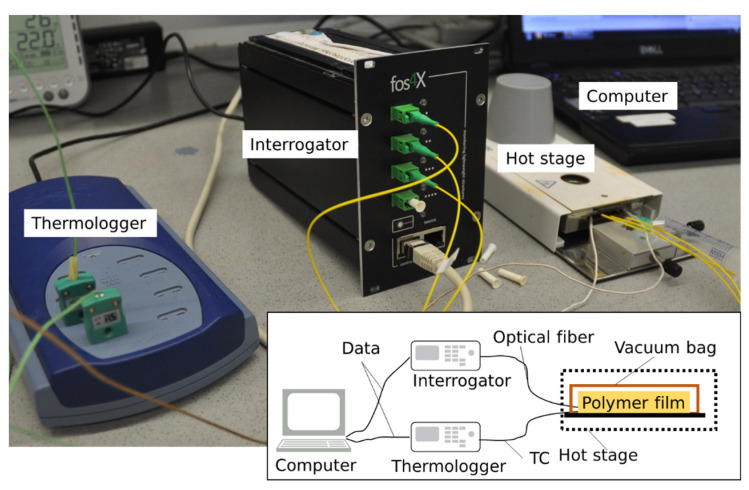
Measurement setup using SMF28e optical fibers and thermocouples of (TC) Class 1 type K.

**Figure 5 sensors-23-03300-f005:**
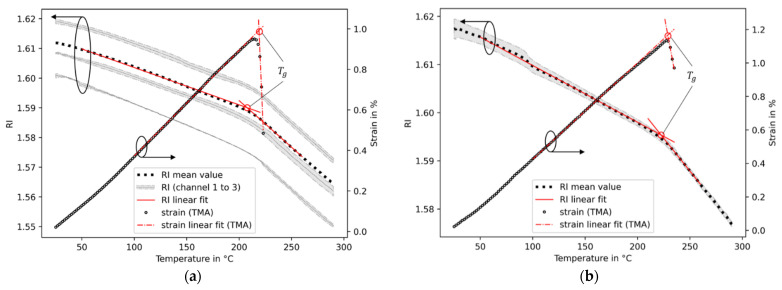
Results of the Fresnel reflectometer refractive index (RI) measurements and thermal strains derived from TMA of PEI (**a**) and PES (**b**), including glass transition temperature T_g_ derived from the intersection points. The grey areas refer to the standard deviations of the corresponding curves.

**Figure 6 sensors-23-03300-f006:**
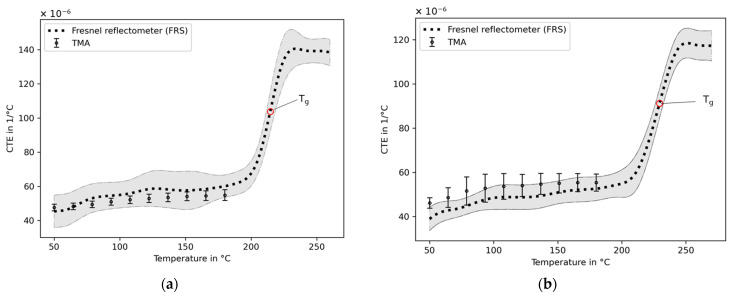
Temperature-dependent CTEs derived from the local derivatives of the refractive index measurements and TMA results of PEI (**a**) and PES (**b**), including the glass transition temperature Tg calculated from the inflection points.

**Figure 7 sensors-23-03300-f007:**
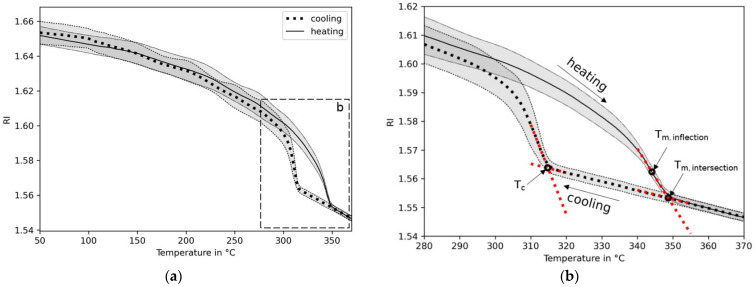
Temperature-dependent refractive indices (RI) derived from the FRS of PEEK (**a**) and focusing on the differences in their paths of the cooling and heating process, including the crystallization (Tc) and melting temperature (Tm, inflection; Tm, intersection) (**b**). The cooling rate displayed is 5 °C/min.

**Figure 8 sensors-23-03300-f008:**
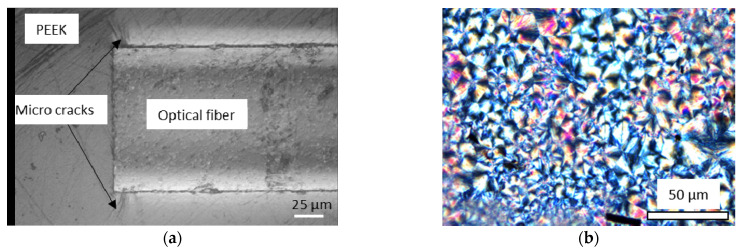
Microsection of the optical fiber embedded into the PEEK polymer (**a**) and PEEK in a petrographic microscope at visible light showing the crystal structure (**b**).

**Figure 9 sensors-23-03300-f009:**
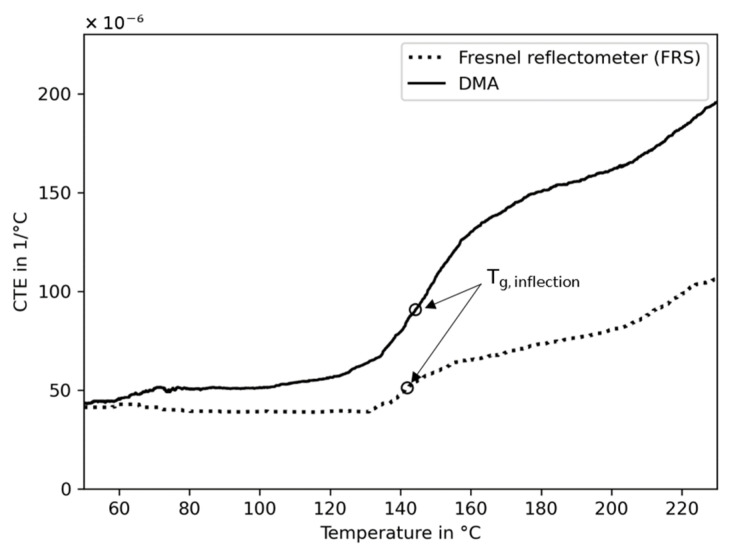
Coefficients of thermal expansion (CTE) and glass transition inflection points (Tg inflection) of PEEK derived from thermal strain measurements by using a calibrated DMA and from the refractive index by using the Fresnel reflectometer sensor (FRS).

**Figure 10 sensors-23-03300-f010:**
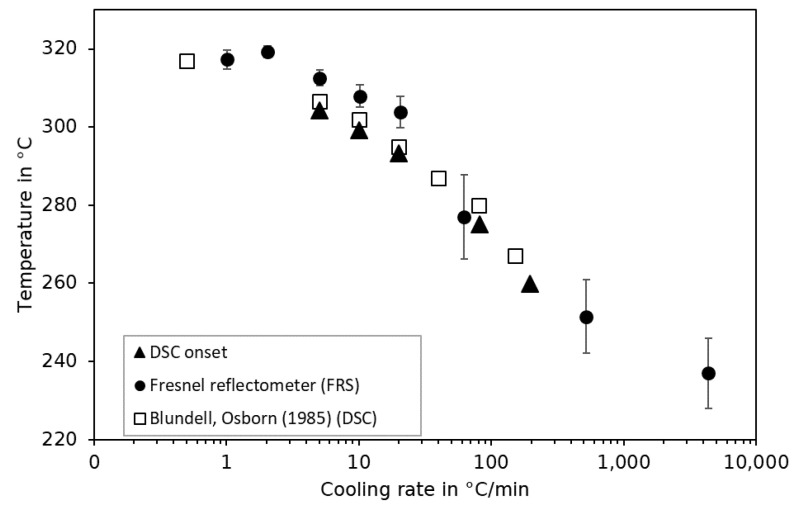
Influence of the crystallization temperature on the cooling rate derived from the DSC and FRS and from the work of Blundell and Osborn (1985) [40].

**Figure 11 sensors-23-03300-f011:**
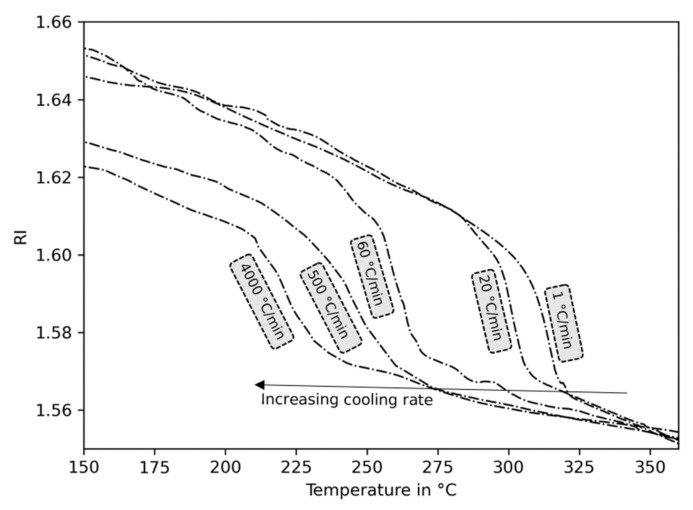
Refractive index (RI) of PEEK at different cooling rates.

**Figure 12 sensors-23-03300-f012:**
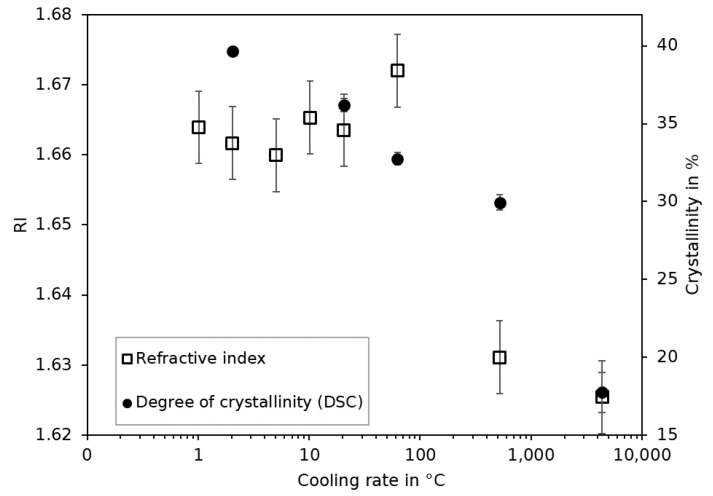
Refractive index (RI) and degree of crystallinity over the cooling rate.

**Table 1 sensors-23-03300-t001:** Linear CTEs derived from the Fresnel reflectometer below and above the glass transition temperature (Tg) and TMA below Tg.

Method	$\mathrm{PEI}\;\left[\times 10^{-6}\;{}^{{}^{\circ}}C^{-1}\right]$	$\mathrm{PES}\;\left[\times 10^{-6}\;{}^{{}^{\circ}}C^{-1}\right]$
Fresnel reflectometer (below Tg)	54.6 ± 4.3	50.9 ± 3.8
TMA (below Tg)	51.9 ± 2.6	56.9 ± 3.9
Fresnel reflectometer (above Tg)	146.3 ± 3.3	124.1 ± 4.4

**Table 2 sensors-23-03300-t002:** Summary of glass transition temperatures.

Method	$\mathrm{PEI}\;\left[{}^{{}^{\circ}}C\right]$	$\mathrm{PES}\;\left[{}^{{}^{\circ}}C\right]$
Fresnel reflectometer (FRS), intersection	207.6 ± 3.3	222.0 ± 3.3
Fresnel reflectometer (FRS), inflection	214.3 ± 1.2	229.4 ± 2.4
TMA, intersection	219.1 ± 1.8	230.1 ± 5.1
DSC, inflection	217.8 ± 0.2	228.5 ± 0.2

**Table 3 sensors-23-03300-t003:** Summary of the transition temperatures of PEEK.

Method	$\mathrm{Glass}\;\mathrm{Transition}\;\mathrm{Temperature}\;\left[{}^{{}^{\circ}}C\right]$	$\mathrm{Melting}\;\mathrm{Temperature}\;\left[{}^{{}^{\circ}}C\right]$
Fresnel reflectometer, inflection	141.9	344.6 ± 1.7
Fresnel reflectometer, intersection	-	346.3 ± 2.5
DMA, inflection	144.2	-
DSC, inflection	144.4 ± 1.5	-
DSC, peak heat flow	-	339.7 ± 1.8

## Data Availability

All data that were generated or that appeared in this study are available upon request by contacting the corresponding author.

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
