# Peer review of "An Edge-Filtered Optical Fiber Interrogator for Thermoplastic Polymer Analysis"

_sensors, 2023, doi:10.3390/s23063300_

Round 1

Reviewer 1 Report

1. Please highlight the creative point and the major finding in abstract. Some sentences in abstract can be deleted. i.e., "No successful correlation...".

The expression in abstract should be in scientific manner. i.e., "... not only..." should be improved.

2. The structure of introduction should be improved. The current drawbacks should be declared. The major work should be described in the last paragraph.

3. Bothe "optical fiber" and "fiber optic" have been used to describe the same thing. Please make them in accordance. "optical fiber" is suggested. 

4. Please provide the physical photo of the testing device in Fig.4.

5. It is suggested that 3.1 should be moved to 2.3.

6. Please explain this sentence "This effect is assumed to increase the lower the temperatures are" in line 316.

7. Please declare the major contribution of the study in Section 5 conclusion part. The current version should be revised.

Reviewer 2 Report

The authors have done a good job and are well presented in the paper. Probably, the choice of the particular materials, namely, the polyetherimide, and the polyethersulfone, but also the polyether ether ketone, investigated should be better emphasized. How much the obtained results be generalized to other groups of related materials?

Reviewer 3 Report

The article is about an edge-filtered fiber optic interrogator for thermoplastic polymer analysis. From the scientific point of view, the authors provide an interesting and well-written article with a good scientific level. The explanations are good. The survey of the literature sources was made with correctly placed references. The obtained results are clearly described and given in tables and figures.

There are only a few remarks/questions to be clarified by the authors, namely:

1.     I don't see a noticeable difference between figures 5 (a) and 5 (b);

2.     It would be interesting for the authors to add in figure 4, the photograph of  the experimental setup
